

# Effect of increased temperature on carbon and nitrogen uptake of two intertidal foraminifera (*Ammonia tepida* and *Haynesina germanica*)

Julia Wukovits[1], Annekatrin J. Enge[1], Wolfgang Wanek[2], Margarete Watzka[2], and Petra Heinz[1]

[1]University of Vienna, Department of Palaeontology, Vienna, Austria
[2]University of Vienna, Department of Microbiology and Ecosystem Science, Terrestrial Ecosystem Research, Vienna, Austria
*Correspondence to:* Julia Wukovits (julia.wukovits@univie.ac.at)

**Abstract.** Benthic foraminifera are highly abundant heterotrophic protists in marine sediments, but intertidal communities are expected to undergo future changes. Environmental changes can exceed the tolerance limits of intertidal species causing a shift in species composition which might result in altered nutrient fluxes. Factors limiting the abundance of specific foraminiferal species can be temperature related stress tolerance or food source processing efficiency. In this study, we performed a laboratory

feeding experiment on *Ammonia tepida* and *Haynesina germanica*, two dominant foraminiferal species of the German Wadden Sea/Friedrichskoog, to test the effect of temperature on phytodetritus ingestion. The specimens were fed with $^{13}$C and $^{15}$N labelled freeze dried *Dunaliella tertiolecta* (green algae) at the start of the experiment and were incubated at 20°C, 25°C, and 30°C respectively. Dual labelling was applied to observe potential temperature effects on the relation of phytodetrital carbon and nitrogen retention. Samples were taken over a period of two weeks. Foraminiferal cytoplasm was isotopically analysed to

investigate differences in carbon and nitrogen uptake derived from the food source. Both species showed a positive response to the provided food source, but carbon uptake rates of *A. tepida* were 10-fold higher compared to those of *H. germanica*. Increased temperatures had a far stronger impact on carbon uptake of *H. germanica* than on *A. tepida*. A distinct increase in levels of phytodetrital derived nitrogen (compared to more steady carbon levels) could be observed over the course of the experiment. The results suggest that higher temperatures have a significant negative effect on the carbon exploitation of H.

germanica. For *A. tepida*, higher carbon uptake rates and the enhanced tolerance range for higher temperatures could outline an advantage in warmer periods, if the main food source consists of chlorophyte phytodetritus. These conditions are likely to impact nutrient fluxes in *A. tepida*/*H. germanica* associations.

## 1 Introduction

The intertidal is an extreme environment, exposed to intense seasonal and diurnal fluctuations in temperature, challenging

the physiological limits of benthic species, for instance foraminifera. Foraminifera are marine heterotrophic protists with a common worldwide occurrence in extant and fossil communities. Along with future environmental changes, extant coastal foraminiferal communities are expected to show changes in assemblage structures (Schafer et al., 1996; Culver and Buzas, 1995) since some intertidal species exhibit a fast response to rapid changes in their environment (e.g. warming). Temperature



affects physiological performances, resulting in competitive drawbacks altering community structures and leading to shifts in nutrient fluxes and ecosystem balance (Brown et al., 2004; Allen et al., 2005; Petchey et al., 2010; O'Connor et al., 2009; Yvon-Durocher et al., 2010).

In intertidal mudflats, smaller benthic foraminifera can contribute up to 80% of the protist biomass (Lei et al., 2014) and are

considered an important element of the food web (Lipps and Valentine, 1970; Buzas, 1978; Buzas and Carle, 1979; Nomaki et al., 2008). But the allocation of their trophic role still lacks defined considerations. Due to their high abundances and substantial carbon incorporation, it is assumed that they play a major role in the carbon cycle of these environments (Moodley et al., 2000). *Haynesina germanica* and *Ammonia tepida* often co-occur with high abundances in intertidal sediments of the temperate zone. The dominance of these two species shows seasonal fluctuations and can be location specific (Murray, 2014;

Alve and Murray, 1994; Debenay et al., 2006; du Chatelet et al., 2009). Factors controlling these community shifts still need to be specified and might include differential sensitivity to general disturbances, differences in food preferences, or variations in physiological limits to physical stress.

Temperature has been proven to play a major role in reproduction, growth and respiration rates of intertidal foraminifera (Bradshaw, 1955, 1957, 1961; Lee and Muller, 1973; Haynert and Schönfeld, 2014; Cesbron et al., 2016). Elevated temperatures

around 35°C were reported to increase the expression of stress proteins in *A. tepida* (Heinz et al., 2012) and represent the range of minimum respiratory activity, which subsequently drops until the lethal point of 45°C (Bradshaw, 1961). There is still a lack of data concerning physiological effects of temperature on *H. germanica*, though there is evidence about lower environmental temperatures (12°C) being most supportive for its reproduction. (Goldstein and Alve, 2011). However, possible restraints or advances of temperature effects on phytodetritus uptake have not been evaluated yet. Foraminiferal food sources

include microalgae, phytodetritus or bacteria (Muller and Lee, 1969; Lee et al., 1966; Goldstein and Corliss, 1994), which are prevalent elements of the intertidal POM (particulate organic matter) pool. Distributions of POM types or microalgae groups are often used to correlate foraminiferal abundances and to relate the availability of different POM sources with population dynamics or food preferences (Alve and Murray, 1994, 2001; du Chatelet et al., 2009; Hayward et al., 1996; Murray and Alve, 2000; Ward et al., 2003; Topping et al., 2006; de Nooijer, 2007; Diz and Francés, 2008; Goineau et al., 2012; Papaspyrou et al.,

2013; Melis and Covelli, 2013). microalgae are suggested to represent the preferred food source of *A. tepida*, in particular over bacteria (Pascal et al., 2008b) , while *H. germanica* possesses a mechanism to efficiently feed on diatoms (Austin et al., 2005). Foraminifera accumulate their food with a pseudopodial network to ingest and transport food particles to the endoplasm (c.f. (Goldstein and Corliss, 1994; Bowser et al., 1992), which is often protected by an inorganic (e.g. calcareous) shell. At this, their ingestion rates of algae or phytodetritus are comparable to bacterial assimilation rates of detrital carbon (Moodley et al.,

30   2000, 2002).

Although there exist abundant data sources about food derived carbon in foraminiferal cytoplasm, information about nitrogen remains scarce. Research about coupling of food derived carbon and nitrogen in foraminifera is limited to an in situ study in the bathyal of the Arabian Sea (Enge et al., 2016), where foraminifera play an important role in benthic carbon fluxes (Enge et al., 2014). Tidal flats function as both an important source and sink for nutrients (Joye et al., 2009) and represent essential ecolog-

ical components of earth?s marine systems. The abundant intertidal primary producers like microalgae or diatoms control C



and N flows of the sediments (Cook et al., 2004), with foraminifera being an important consumer of their biomass. Regarding the high abundances of foraminifera in benthic communities, significant effects of temperature on their food processing could potentially influence intertidal nutrient fluxes, in particular if they relate to foraminiferal carbon and nitrogen coupling. This study therefore aimed to investigate the effects of temperature on the food exploitation and on C and N cycling of *H. ger-*
*manica* and *A. tepida*. The response to an artificially produced phytodetrital food source (*Dunaliella tertiolecta*, Chlorophyta) was tested under three temperature regimes. *Dunaliella tertiolecta* has been identified as a valuable food source for *A. tepida* (Pascal et al., 2008a), but was not tested on *H. germanica* so far. This study therefore also tests the question which of the two species shows a more efficient response to chlorophyte detritus in a mesocosm setting. The laboratory feeding experiment was performed in incubation chambers, adjusted to 20°C, 25°C and 30°C. To obtain direct estimates of the uptake of carbon and
nitrogen, phytodetritus was labelled with stable isotopes ($^{13}$C, $^{15}$N). This method has provided important data on the in situ feeding behaviour of foraminifera in various environments (Moodley et al., 2002; Enge et al., 2016; Moodley et al., 2000; Middelburg et al., 2000; Witte et al., 2003; Nomaki et al., 2005, 2009, 2011; Sweetman et al., 2009; Enge et al., 2011; Jeffreys et al., 2013). The temperatures chosen for this experiment correspond to experimentally determined values that cover optimum or tolerance ranges for growth and reproduction in laboratory cultures of intertidal foraminifera (Bradshaw, 1957, 1961; Lee
and Muller, 1973). Further, they lie in the range of seasonal and diurnal temperature amplitudes measured on intertidal surface sediments close to the sampling area (Al-Raei et al., 2009). Simulated variations in temperature were assumed to influence the food uptake efficiency of the species, due to potential temperature stress. The amount of phytodetrital carbon (pC) and nitrogen (pN) uptake should reveal information about the nutrient processing potential of the two species. Simultaneous detection of both stable isotopes allows to determine the ratio in which pC relative to pN is retained in bulk foraminiferal cytoplasm over
time. This helps to interpret nutritional demands (Enge et al., 2016; Evrard et al., 2010; Hunter et al., 2012) and temperature influences might show in imbalances of these ratios between treatments. Benthic foraminifera are used in the assessment of (paleo-) environmental data. Studies aiming on the development of new foraminiferal proxies (e.g. for organic matter accumulation or physical parameters) generally apply statistical analysis to field surveys. There is however a comparably low amount of biological or ecological data available. This study offers distinct observations on the effect of an altered environmental
condition (temperature) on food resource exploitation on the level of the cytoplasmatic balance of food derived carbon and nitrogen. The aim is to support definitions of the role of benthic foraminifera in intertidal carbon and nitrogen fluxes with respect to warming events and changes in POM availability.

## 2   Material and Methods

### 2.1   Sampling site & Material collection

Surface sediment was taken on 24th April 2014 during low tide in the intertidal mudflat of the German Wadden Sea near Friedrichskoog (Germany). Water temperature and salinity at the sampling site were 19.9°C and 31 psu (practical salinity units). Sediment was collected and sieved at the sampling site through 500 $\mu$m and 63 $\mu$m meshes to remove larger meiofauna and organic particles. In the laboratory, samples were sieved again to obtain foraminiferal specimen in the size fraction of



125 - 355 $\mu$m. The sediment contained high abundances of *Ammonia tepida* and *Haynesina germanica* individuals, which were picked and collected for further processing. Living individuals were identified under the microscope regarding intact protoplasma and particle accumulation around the aperture (Moodley et al., 2002; Nomaki et al., 2005; Moodley et al., 1997; Nomaki et al., 2006). Care was taken to achieve a homogenous size distribution within specimens in each replicate. Samples of each species were kept separately in crystallising dishes and fed regularly with living *Dunaliella tertiolecta* until the start of the experiment.

### 2.1.1 Production of $^{13}$C and $^{15}$N enriched phytodetritus

At the start of the experiment, lyophilized powder of *Dunaliella tertiolecta* was used to simulate a phytodetritus pulse. The chlorophyte was grown in f/2 medium (Guillard and Ryther, 1962; Guillard, 1975) enriched with 98 atom % $^{13}$C (NaH$^{13}$CO$_3$, Sigma-Aldrich) and 98 atom % $^{15}$N (Na$^{15}$NO$_3$, Sigma-Aldrich) to final concentrations of 1.5 mmol L-1 NaH$^{13}$CO$_3$ and 0.44 mmol L-1 Na$^{15}$NO$_3$. Algal cultures were kept within incubators (T = 20°C; dark: light = 16:8). After 20 days, they were harvested by centrifugation (800 G; 10 min) and rinsed three times in simplified artificial seawater (ASW, compare preparation in Enge et al. (2011) to remove unassimilated isotope tracer. The hereby obtained algal slurry was shock frozen with liquid nitrogen and lyophilized at -55°C and 0.180 mbar for 6 days.

### 2.1.2 Experimental setup

Separate time series for *A. tepida* and *H. germanica* were performed at three temperatures (20°C, 25°C, 30°C) in triplicate. Additionally, background samples of untreated specimens were taken to obtain natural abundance values for $^{13}$C/$^{12}$C and $^{15}$N/$^{14}$N and initial cytoplasmatic total organic carbon (C) and nitrogen (N) content. Specimens were transferred into 72 experimental dishes (150 individuals of *A. tepida*/170 individuals of *H. germanica* per dish), each containing 280 mL modified synthetic seawater (SSW; (Kester et al., 1967; Dickson and Goyet, 1994), adjusted to a pH of 8.10 and a salinity of 32 psu. The dishes were incubated at a light: dark = 12:12 hrs cycle (type ST 2 POL-ECO Aparatura incubation chambers). After three days of acclimation, the labelled algal diet was added (396.36 $\pm$ 45.57 mg C m$^{-2}$; 40.26 $\pm$ 9.43 mg N m$^2$). No further feed amendment was provided throughout the experiment. Algal particles were visible as a green layer on the bottom of the experimental vessels until the end of the experiment. Oxygen, salinity and pH were kept constant at optimum levels during cultivation. Experimental dishes were sealed airtight and opened after two days to avoid hypoxia. Both species were subsampled to obtain data of phytodetritus processing on the 2nd, 4th, 7th and 14th day of the experiment.

### 2.1.3 Sample preparation

After opening the experimental dishes on the 2nd day, water samples (aliquots of 50 mL) were taken from the *H. germanica* series. They were transferred to 50 mL headspace vials. Some drops of HgCl$_3$ were added to stop respiratory activities and biological production of CO$_2$ (Kroopnick et al., 1972). The vials were sealed airtight and stored at 4°C. To determine $^{13}$C of the dissolved inorganic carbon (DIC) in the samples, 12 mL vials were flushed with He, filled with 0.5 mL 85% H$_3$PO$_4$ and 2



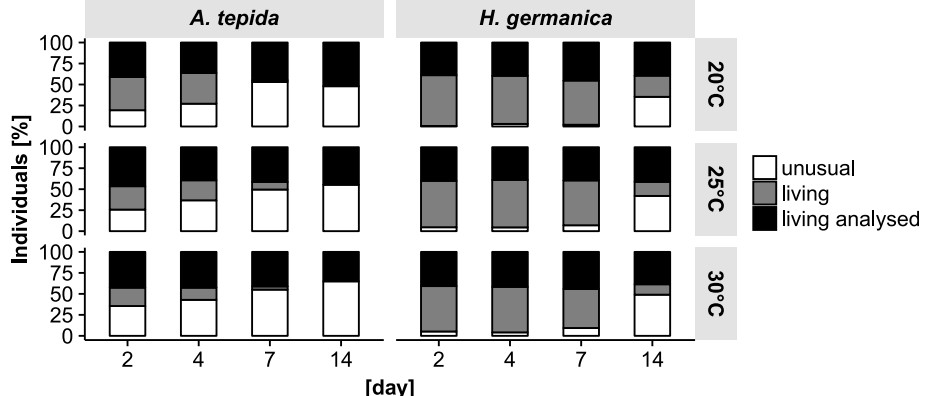

**Figure 1.** Relative amounts of individuals per sampling day and temperature treatment. The terms unusual and intact refer to the cytoplasmatic appearance of the specimen, additionally the fraction of analysed specimen is shown. Unusual cytoplasm includes individuals with patchy distribution of cytoplasm within the test or an unusual coloration. Individuals with intact cytoplasm includes former live specimens as described in the text.

mL of these water samples, sealed airtight and stored to equilibrate for 48 h (Li et al., 2007; Taipale and Sonninen, 2009). Foraminifera were removed from the experimental dishes and frozen at -20°C to stop metabolic activities. The foraminifera were cleaned from adhering particles with a hair brush. Further on, the organisms were carefully washed in simplified ASW. An amount of 50 (*A. tepida*) or 60 (*H. germanica*) individuals met the optimum range of 0.7 - 1.0 mg cytoplasmatic dry weight, necessary for isotope and elemental analysis. Only foraminifera meeting the criteria for live specimens as described above were prepared for analysis. Figure 1 shows relative amounts of cytoplasm conditions and analysed individuals on the respective sampling days. Specimens were transferred to tin capsules, dried at 50°C, decalcified and dried in a final drying step for three days. All glassware used for preparation was combusted at 500°C for 5h, picking tools and tin capsules were cleaned in a solution of dichloromethane ($CH_2Cl_2$) and methanol ($CH_4O$) (1:1, v:v).

## 2.1.4  Sample analyses

Foraminiferal samples and water samples were analysed at the Stable Isotope Laboratory at the University of Vienna for Environmental Research (SILVER). Phytodetrital (dry wheight) and foraminiferal content of organic carbon or nitrogen and ratios of $^{13}C/^{12}C$ and $^{15}N/^{14}N$ were determined with an Isotope Ratio Mass Spectrometer (IRMS; DeltaPLUS, Thermo Finnigan) coupled with an interface (ConFlo III, Thermo Finnigan) to an elemental analyzer (EA 1110, CE Instruments). $\delta^{13}C$ of dissolved inorganic carbon (DIC-$\delta^{13}C$) was measured after release as $CO_2$ by $H_3PO_4$ addition in the headspace (headspace gas sampler: GasBench II, Thermo Fisher) of the prepared samples using an IRMS (Delta Advantage V, Thermo Fisher). Atom%



of the samples were derived from isotope ratio data and were calculated using the Vienna PeeDee Belemnite standard for C (RVPDB = 0.0112372) and atmospheric nitrogen for N (RatmN = 0.0036765), where X is $^{13}$C or $^{15}$N:

$$atom\% = \frac{100 * R_{\text{standard}} * (\frac{\delta X_{\text{sample}}}{1000} + 1)}{1 + R_{\text{standard}} * (\frac{\delta X_{\text{sample}}}{1000} + 1)} \qquad (1)$$

Net uptake (uptake into the foraminiferal cell, excluding released amounts, hereafter referred to as 'uptake') of phytodetrital carbon and nitrogen in foraminiferal cytoplasm was calculated by determining the excess (E) of isotope content within the samples against the natural abundance of the isotopes in the foraminiferal cytoplasm (Middelburg et al., 2000):

$$E = \frac{atom X_{\text{sample}} - atom X_{\text{background}}}{100} \qquad (2)$$

where X is $^{13}$C or $^{15}$N. Excess and total organic carbon and nitrogen (C, N, biomass normalised, per mg sample weight, or per individual) were used to calculate the amount of incorporated isotope $I_{\text{iso}}$ [$\mu$g mg$^{-1}$] or [$\mu$g ind$^{-1}$] = E x C (N) [$\mu$g mg$^{-1}$] or [$\mu$g ind$^{-1}$], to obtain the amount of phytodetrital carbon (pC [$\mu$g mg$^{-1}$] or [$\mu$g ind$^{-1}$]) and nitrogen (pN [$\mu$g mg$^{-1}$] or [$\mu$g ind$^{-1}$]) within the foraminiferal cytoplasm (Hunter et al., 2012):

$$pX = \frac{I_{\text{iso}}}{\frac{atom\% X_{\text{phyto}}}{100}} \qquad (3)$$

For a better comparison of the variation in uptake dynamics between the two species, the uptake of labelled partiles was time normalized to obtain uptake rates of phytodetrital carbon and nitrogen (ng mg$^{-1}$ h$^{-1}$). Exponential decay functions (y = a (-b*x)) were applied with least squares global curve fitting for carbon uptake rates. The steepness of the decrease b (high b value = fast decrease) indicates whether there is a fast or slow drop in the uptake rates within the related samples over time. An F-test was carried out with mean values of treatment curves to compare the nonlinear regression models (exponential decrease) for uptake rates of phytodetrital carbon and evaluate if a single model can be fitted for the two species at 20°C. The same procedure was used to prove if uptake rates of phytodetrital carbon vary withn species between the three temperature treatments. Welch's t-test was used to detect differences of C:N ratios, pC:pN ratios and cytoplasmatic C and N content between species. This test was chosen because it is recommended for sample sizes < 10 (McDonald, 2014) and it shows only slight variations in power compared to Student's t-test (e.g. (Moser and Stevens, 1992; Ruxton, 2006) and can still be applied in case of unequal variances. To compare within species differences, a two-way ANOVA was carried out beforehand with time and temperature as independent factors and pC, pN, C and N as dependent variables. Depending on the result, pairwise t-testing was used to compare groups when no interactive effect was observed. A parametric ANOVA was carried out when interactions were present, followed by a Tukey HSD test when significances were detected. Testing for normality in a sample size n = 3 will not give a reliable estimation of the true distribution within the population. Obtaining replication of n > 30 would have expanded the pre-experimental cultivation and isolation period of the specimen and therefore enhanced chances of unnatural reactions





to the experimental treatments. The experimental setup was designed to assume a normal distribution of the feeding behaviour within the population of the collected foraminifera. Anyway, parametric ANOVA is reported to be robust against violation of the normal distribution (Schmider et al., 2010) and was chosen over a non-parametric alternative (Kruskal-Wallis ANOVA), which compensates for nonnormal distributions. Homogenity of variances was tested using Fligner-Killeen's test which can be

applied when population means are not known (Conover et al., 1981) and can overcome some problems with small sample sizes (Wasserstein and Boyer Jr, 1991). The statistical analysis applied here were meant to emphasise the visualizations (graphical depictions) of the findings of this study. The applyied tests were chosen to suit and repsresent the data in the most appropriate way (to retain robustness with respect to the design of the study). Graphs and data analysis were done using R (R Development Core Team, 2008) via Rstudio (RStudio Team, 2015) and the packages ggplot 2 (Wickham, 2009) nlstools (Florent Baty et al.,

2015) and plyr (Wickham, 2011).

## 3  Results

### 3.1  Carbon processing

Remarkably elevated $^{13}$C values of the foraminiferal samples demonstrated the strong response to the labelled food source (Table 1). Temperature had a significant impact on pC levels in *H. germanica* (Table 2 and Fig 2). An interactive effect of

time and temperature caused a different time related processing of pC at 20°C compared to 25° and 30°C (Table 2 and Fig 2). The content of pC in *H. germanica* at 20°C was significantly higher than at 25°C and 30°C, on each day of data collection (p< 0.05; no significant difference in 25°C and 30°C samples). Additionally, values for DIC-$^{13}$C measured in water samples at elevated temperatures were significantly higher than those from the 20°C approach (Table 1). Time related variations in cytoplasmatic C (biomass normalised) in *H. germanica* specimens were also temperature dependent (Table 2). In contrast,

there was no relation between time and temperature concerning the pC content of *A. tepida*. Levels of pC in *A. tepida* were significantly increased at 25°C (25°C 20°C: p = 0.015, 25°C - 30°C: p< 0.001, pairwise t-testing with pooled SDs, since no significance of time effects), revealing a higher optimum grazing temperature for *A. tepida* compared to *H. germanica* (pC, Fig 2). Temperature had no effect on biomass C in *A. tepida*, while there were time dependent variations in biomass C (Table 2). These appeared at the start and end of the experiment but no significant influence by temperature treatments on the other days.

Individual cytoplasmatic C content [$\mu$g ind$^{-1}$] was different between the two species and proportions of individual uptake of pC to individual C content showed similar proportions like biomass related carbon values (Fig 3).

Carbon uptake rates showed an exponential decrease with time for both species (Fig 4), with a very good fit of curves except for *A. tepida* at 25°C where a relatively low determination coefficient $R^2$ was observed (Table 3). Mean uptake rates display a highly significant difference between species (F-test: F = 33.74, p = 0.029). *H. germanica* showed a much lower efficiency in

the uptake of phytodetritus derived from *D. tertiolecta* compared to *A. tepida*. Uptake rates in *A. tepida* can be pooled within a single function (F-test; F = 7.664, p = 0.062). Temperature effect causes a strong variation in rates for *H. germanica* (F-test; F = 24.97, p = 0.012). The rates of *A. tepida* showed a slightly faster decrease at 20°C relative to the higher temperatures and



**Table 1.** Delta values for carbon and nitrogen isotopes. Natural $\delta^{13}$C and $\delta^{13}$N values of unlabelled background samples (BG), of isotopically enriched samples of food (*D. tertiolecta*) and foraminifera, and DIC-$\delta^{13}$C within water samples at the 2nd day of the cultivation period (standard deviation in parenthesis; letters denote Tukey Grouping of significant differences in DIC-$\delta^{13}$C between temperatures; n.d. = no data).

| | | $\delta^{13}$C [‰] | $\delta^{15}$N [‰] | DIC $\delta^{13}$C |
|---|---|---|---|---|
| *D. tertiolecta* | BG | -18.3 | 16.2 | n.d. |
| | labeled | 30267 | 213298 | n.d. |
| *A. tepida* | BG | -13.9 (± 0.2) | 13.4 (± 0.6) | n.d. |
| | 20°C | 1742 (± 216) | 7277 (± 1203) | n.d. |
| | 25°C | 1269 (± 318) | 5758 (± 1426) | n.d. |
| | 30°C | 707 (± 122) | 4109 (± 320) | n.d. |
| *H. germanica* | BG | -13.7 (± 0.7) | 11.7 (± 0.4) | n.d. |
| | 20°C | 248 (± 26) | 1087 (± 71) | 1845 (± 76) **a** |
| | 25°C | 118 (± 20) | 784 (± 86) | 2186 (± 60) **b** |
| | 30°C | 116 (± 5) | 879 (± 29) | 2214 (± 66) **b** |

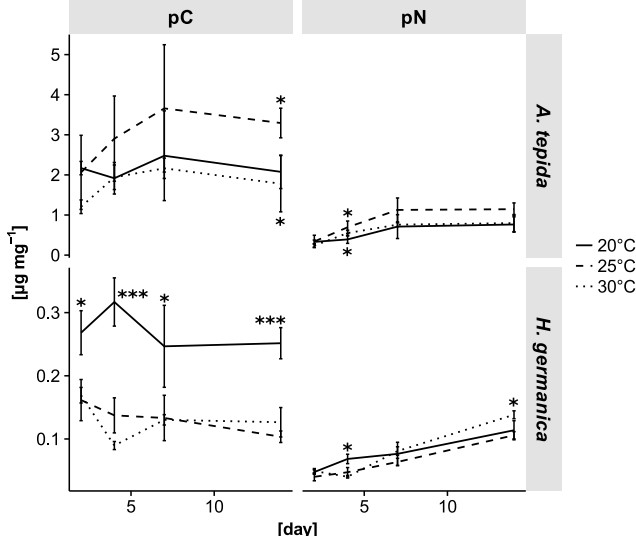

**Figure 2.** Phytodetrital derived carbon and nitrogen. Biomass normalized pC and pN of *A. tepida* and *H.germanica* at 20°C, 25°C, and 30°C. Error bars denote standard deviation, stars indicate significance (*p <0.050, **p <0.010, ***p <0.001, TukeyHSD), for statistics details see text.

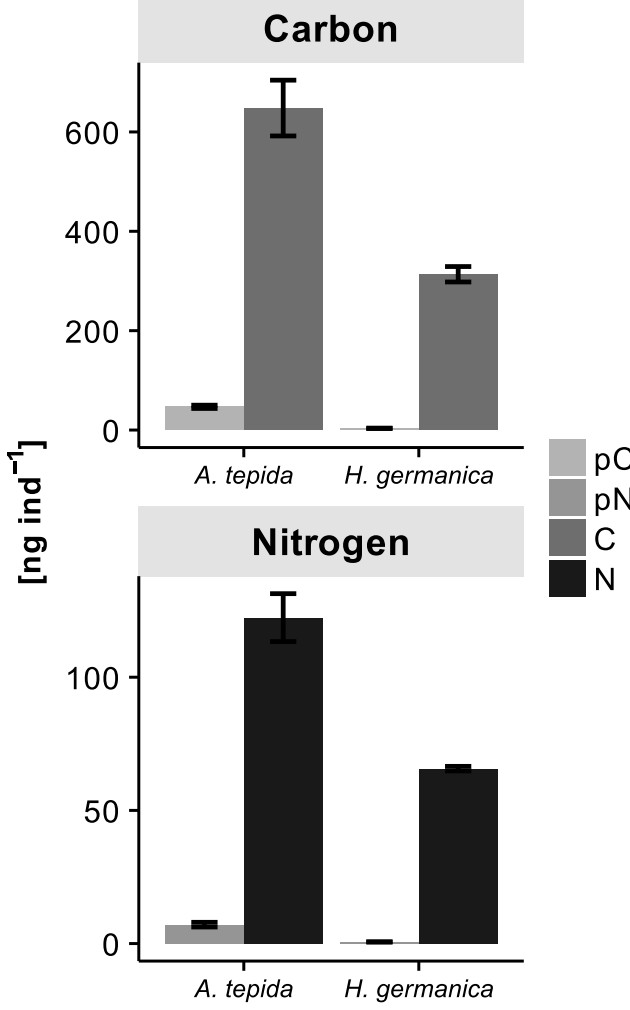

**Figure 3.** Cytoplasmatic carbon (C) and nitrogen (N) content and pC or pN uptake per foraminiferal individual. Data represents values for day 2 of the 20°C approach. Error bars denote standard deviation.

the highest deviation (S) of 21.247 ng mg⁻¹ h⁻¹ from average uptake rates at 25°C. In general, *H. germanica* exhibited a faster drop in the rates at the higher temperatures (c.f. b, Table 3).

## 3.2 Nitrogen processing

Same as for the $\delta^{13}$C values, $\delta^{15}$N signatures of all foraminiferal cytoplasm samples showed a strong increase after addition of labelled algae (Table 1). Like in pC, the effects of time and temperature on the pN content of *H. germanica* interacted (Table 2). At day 4a remarkable temperature related variation of pN in *H. germanica* was evident (20°C - 25°C: p = 0.014, 20°C - 30°C:





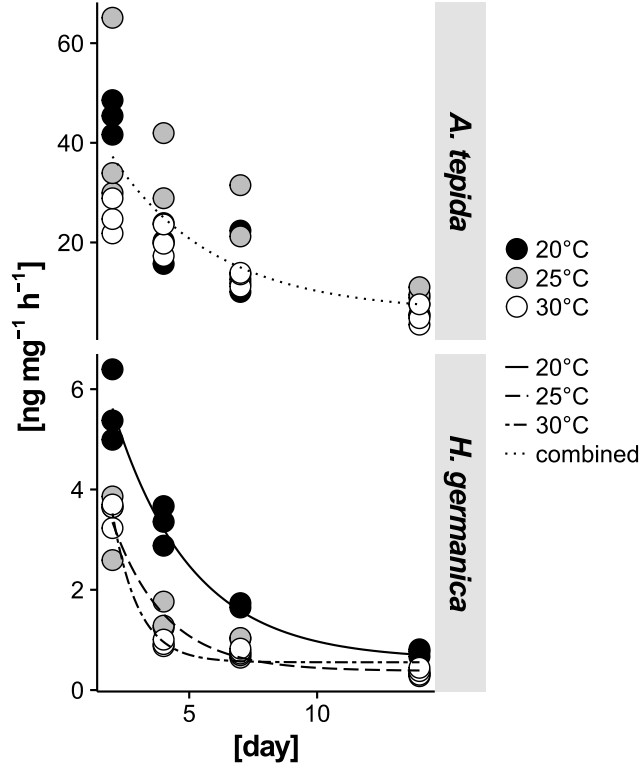

**Figure 4.** Carbon uptake rates. Uptake rates of phytodetrital carbon of *A. tepida* and *H.germanica* at three temperatures. Dots represent the calculated uptake rates derived from isotope data per time, curves show functions of exponential decrease.

5    p = 0.003), as on day 14 (25°C - 30°C: p = 0.017; rest = n.s.). Biomass normalised cytoplasmatic N of *H. germanica* showed diffuse fluctuations at 20°C and 25°C, and differences between temperatures at the second day (20°C - 25°C: p = 0.034; 20°C - 30°C: p = 0.003, rest = n.s.). pN showed less variation with temperature in *A. tepida* (20°C - 25°C: p = 0.041; rest = n.s.). In contrast to the pC values, progressing time caused significant increases in pN content in both species (Fig 2, pN). N changes in *A. tepida* were time dependent and increased at the start of the 25°C and 30°C approach. Analogous to cytoplasmatic C, individual N and pN are proportional to total biomass related values (Fig 3).

     In general, nitrogen uptake rates of *H. germanica* showed a steep decline over time and at all temperatures in contrast to *A.*

5    *tepida* (Fig 5). While *H. germanica* showed the highest uptake rates at 20°C, uptake rates for *A. tepida* were highest at 25°C. At 30°C, uptake rates of phytodetrital nitrogen showed fluctuating patterns for both species. In Ammonia samples, rates at 25°C and 30°C drop after the 4th day. The steepest decrease at 20°C can be found between day 2 and day 4. Uptake rates of nitrogen were about tenfold higher in *A. tepida* compared to *H. germanica*.

     There was a noticeable peak in N uptake on the 4th day at 30°C in both species. In *A. tepida*, this peak describes an increase

10   of the uptake rate followed by a linear decrease. In contrast, all uptake rates showed a rapid drop in *H. germanica* especially at





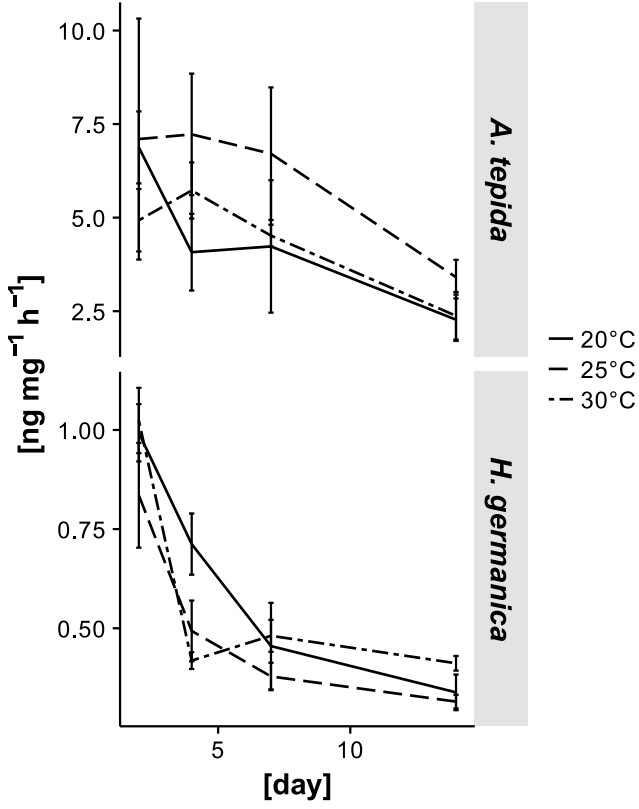

**Figure 5.** Nitrogen uptake rates. Uptake rates of phytodetrital nitrogen by *A. tepida* and *H.germanica* at 20°C, 25°C, and 30°C. Error bars denote standard deviation

the higher temperatures with a recognisable negative peak at 30°C on day 4, and a flattening of nitrogen uptake rates at 25°C and 30°C on day 4.

### 3.3 Relationship of cytoplasmatic and phytodetrital derived carbon and nitrogen content

In *H. germanica* , the amounts of pC and pN converged with increasing time, due to $^{15}$N enrichment and $^{13}$C loss over time (Fig 2). This enrichment of pN was detectable in both species, but was generally steeper in *H. germanica* with an almost equal increase at all temperatures until the end of the experiment. In contrast, the steady rise of pN in *A. tepida* receded after seven days. It is noticeable, that despite the constant enrichment of pN in *H. germanica*, pC levels were lower at higher temperatures. This faster loss of pC at 25°C and 30°C at similar levels of phytodetritus uptake was also reflected in lower pC:pN ratios at higher temperatures (Fig 6). Moreover, the interactive effect of temperature and time on the food uptake of *H. germanica* was also reflected in the decoupling of pC and pN (Table 2 and Fig 7) and the lack of correlation between phytodetrital carbon and nitrogen content in the cytoplasm of *H. germanica* over the course of the experiment (Fig 7). In contrast, pC and pN were





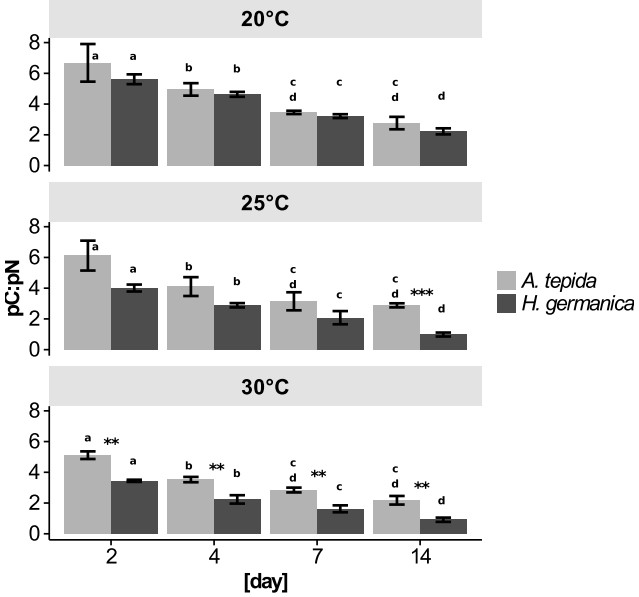

**Figure 6.** Ratios of cytoplasmatic pC:pN ratios. Biomass normalized pC:pN within foraminiferal cytoplasm. Error bars denote standard deviation, stars indicate significant differences of pC:pN ratios between the two species on the respective day (t-test; *p <0.050, **p <0.010; ***p <0.001). Letters show Tukey groupings of variations in pC:pN ratios between sampling days.

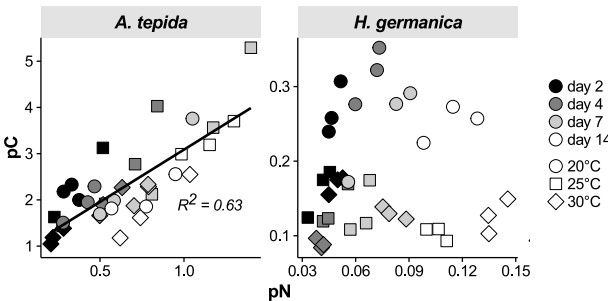

**Figure 7.** Carbon and nitrogen coupling. Phytodetrital derived nitrogen (pN) vs. carbon (pC) of *A. tepida* and *H. germanica* at 20°C (circles), 25°C (squares), and 30°C with linear regression for *A. tepida*.

10  strongly coupled in *A. tepida* and this across the whole time series (Fig 7). Cytoplasmatic C:N ratios were generally similar in both species, with no significant impact of temperature (Table 4).





## 4 Discussion

### 4.1 Effect of temperature on carbon and nitrogen uptake

The metabolism of *H. germanica* was significantly affected by temperature. Higher temperatures reduced the amount of pC in the foraminiferal cytoplasm, while otherwise a slighter effect on pN indicates a lower impact of temperature on the general uptake of algal phytodetritus. The loss of pC (Fig 2) in the warmer environment or the faster decline of uptake rates at 25°C and 30°C (Fig 4 and Table 3) was most likely caused by elevated respiration rates. Accordingly, increased DIC-$^{13}$C content of the experimental medium at elevated temperatures suggests higher respiratory activities (Table 1). Increased temperatures have been reported to increase respiration and food consumption in aquatic herbivores or detrivores and foraminifera (Bradshaw, 1961; Carr and Bruno, 2013), to lower grazing rates with simultaneously increased oxygen consumption (Ferreira et al., 2010), or to raise respiration to cover the costs of maintenance of protein metabolism levels (Whiteley and Faulkner, 2005). In general, DIC-$^{13}$C values in this experiment were very high compared to foraminiferal $^{13}$C. It cannot be excluded that the foraminiferal cultures contained bacteria. Thus, a considerable amount of DIC could originate from phytodetritus remineralisation by microbial activity. Several perturbation- and long term experiments on marine bacteria have also reported increased respiration due to warming (Vázouez-Domínguez et al., 2007; Hoppe et al., 2008; Wohlers et al., 2009). Anyway, the higher temperatures trigger noticeable stress in *H. germanica*, which impacts food uptake efficiency. The interactive time/temperature effect supports this argument, apparent as the higher temperatures caused convergence of pC and pN content compared to the 20°C results, particularly until day 7. This indicates an offset of nutritional ingestion performance in the two warmer approaches. It further suggests a general preference of lower environmental temperatures. Evidently, lower temperatures are more supportive to trigger reproduction in *H. germanica* (Goldstein and Alve, 2011). Interestingly, there is no significant temperature effect on pC between 25°C and 30°C. This implicates a critical threshold for this species between 20°C and 25°C.

In contrast, *A. tepida* showed a trend for optimum uptake of carbon at 25°C and the steadiest carbon uptake rates at 30°C, while the pC and pN values showed similar patterns (Fig 2 and Fig 4). Laboratory experiments on *A. tepida* specimens collected at the Brouage mudflat (France) revealed an optimum temperature of grazing on bacteria at 30°C (Pascal et al., 2008b). This indicates that strains of *A. tepida* feature adaptations to the higher average sea surface temperatures of the Brouage mudflat. In other laboratory experiments (Bradshaw, 1957, 1961) warm temperatures between 25°C and 30°C, offered optimum conditions for reproduction, resulted in higher growth and reproductive rates and a decrease in generation time in laboratory experiments with *A. tepida* specimen. Accordingly, higher temperatures are likely to offer competitive advantages for *A. tepida* over *H. germanica* in terms of food uptake. Considering the high expression of stress proteins at 35°C in *A. tepida* (Heinz et al., 2012) in relation to the here presented results, a critical temperature limit effecting physiologic performances is likely to be found between 30°C and 35°C in this species.

Interestingly, the higher temperature treatments caused an increase of biomass C and N in *H. germanica* samples on the second sampling day, followed by a quick decrease on the fourth day. A similar trend was observed in *A. tepida*. This effect is difficult to interpret and more detailed information about initial food uptake between day 0 and day 2 would be necessary. In summary, keeping up food C and N uptake appears to be a key factor in the strategies of *A. tepida*, according to the strong





30  response to phytodetritus and the low effect of temperature on its uptake. Simultaneously, a high viability of the specimen in culture implies specific environmental adaptations and a strong response and sensitivity to transfer related disturbances (Fig 2). In contrast, *H. germanica* showed a comparably high robustness and overall vitality, but an obviously strong impact of temperature on mechanisms involved in carbon processing. This implicates physiological resource investment for survival by costs of increased metabolic activity.

### 4.2   Relationships between carbon and nitrogen uptake and cytoplasmatic C:N ratios

In both species, the relation of pC to pN (pC:pN ratio) within foraminiferal cytoplasm decreased compared to their phytodetrital C:N ratio of ~10 (Table 4) as time progressed. In theory, a comparably high ratio of detrital derived pC:pN would approach or undercut somatic consumer C:N ratio to retain homeostasis or ensure appropriate nutrition (Cross et al., 2005; Frost et al.,

2005; Sterner and Elser, 2002). The growing content of pN within foraminiferal cytoplasm over time with a remarkable increase of [15]N values (Table 1 and Fig 2) could be explained by the integration of a high amount of food derived nitrogen into amino acids and metabolic consumption of pC. Benthic foraminifera inhabiting Antarctic sediments have been reported to use the major amount of algae derived carbon sources to synthesize nitrogen rich proteins in relation to a lower generation of other products with lower nitrogen content (Rivkin and DeLaca, 1990). Such effects could be responsible for the relatively slight

decrease in nitrogen uptake rates over time compared to the strong decrease in carbon uptake rates (Fig 4 and Fig 5). A strong decline of continued food uptake or degradation induced change of phytodetrital C:N could likewise be responsible for this effect.

Both tested species showed similar demands for food derived carbon and nitrogen, since at moderate temperatures (20°C) pC:pN ratios showed a similar trend over time (Fig 6). Again, minor temperature influences were demonstrated within the

overall linear relationship of pC to pN in *A. tepida* across the samples (Fig 7). In contrast, the rising variation and highly significant difference in pC:pN (lower ratios at higher temperatures, Fig 6) resulting in the diffuse relationship of pC:pN (Fig 7) in *H. germanica* reflect the thermal stress on the level of increased carbon expenditure at steady feeding progress (barely influenced pN levels between treatments). Therefore, sediments with high dominance of *H. germanica* like river inlets or estuaries (Debenay et al., 2006; Murray and Alve, 2000) could possibly experience additional temperature induced nutrient flux

variations, boosting C losses while triggering N retention.

An *in situ* experiment, monitoring population shifts of near shore benthic foraminifera to artificially heated sediments, proposed Migrations to deeper (colder) regions as a response to elevated environmental temperatures (Schafer et al., 1996). Marine species are known to Migrate linked to temperature changes (Parmesan, 2006). To maintain an optimum energy budget, expenses of metabolic maintenance could therefore be compensated by a shift of *H. germanica* populations towards the subtidal.

Foraminiferal C:N ratios are known from bathyal foraminifera of the Sagami Bay and from oxygen minimum zones of the Arabian Sea, where they range from 2.6 - 6.4 (Nomaki et al., 2008, 2011; Enge et al., 2016). In our study, both species showed similar C:N ratios and natural [13]C and [15]N signatures (Table 1 and Table 4), indicating equivalent trophic levels and similar nutritional demands. During the course of the experiment, the temperature stress in *H. germanica* was even noticeable on the level of cytoplasmatic stoichiometry. The remarkable drop of the C:N ratio on the 4th day at 30°C (Table 4) and the subsequent





recovery and rapid increase of pC and pN after the 4th day (Fig 2) could denote a metabolic adaptation process. However, the strong temperature influence on carbon uptake persisted.

In *A. tepida*, the occasionally significant (at 25°C) fluctuations in the C:N ratio can be related to the high uptake of phytodetritus containing high C:N ratios and therefore an influence of food source C:N on the grazer C:N (Fig 3 and Table 4). Generally, considerations of C:N ratios and intracellular pC/pN coupling also reflect the lower effects of high temperature exposure on *A. tepida* and the high impact on the feeding behaviour of *H. germanica*. These observations reflect the different strategies of the two coexisting intertidal foraminiferal species on the level of nutrient balance. It further implies shifts in population distributions of *A. tepida* and *H. germanica* at events of persisting food source and/or temperature changes. This study verifies a strong involvement of foraminifera in the turnover of intertidal POM (high uptake rates, compare also (Moodley et al., 2000).

Depending on POM source or dominant foraminiferal species, this turnover can be strongly influenced by increased temperature, causing a decoupling of carbon and nitrogen cycling. However, when relating these findings to natural environments, considerations of cascading effects of temperature change e.g. switches in trophic conditions and related shifts in nutrient coupling of primary producers have to be included.

### 4.3  Species specific food uptake

The very high uptake rates and levels of phytodetrital carbon and nitrogen in *Ammonia tepida* (day 2 ~14% pC:C; ~10% pN:N, comp. Fig 3 and Fig 4) indicate a strong response to the food source, in contrast to *H. germanica* which showed a clear, but much lower uptake (day 2: ~2% pC:C; ~1.5% pN:N). The observed decline of carbon uptake rates by time in both species suggests the highest uptake within the first two days of the experiment and indicates a fast processing of the phytodetritus source. During an in situ experiment, Moodley et al. (2000) also observed a rapid response of Ammonia sp. to green algae

(*Chlorella* sp.) within hours, together with the much weaker ingestion of such food source by Haynesina sp.

The quick slowdown of food uptake rates could indicate levels of saturation especially after a strong response following the food pulse. In addition, cellular release of label is likely to exceed uptake as time progresses. Other reasons for the decrease in food uptake with increasing time (reflected in decreasing uptake rates of C and N as well as in pC and pN) could be the stage of phytodetrital decay or decreasing food availability at the subsequent sampling days. However, a visible layer of phytodetritus

particles persisted until the end of the experiment. The influence of food availability on feeding, where high food concentrations support increased feeding rates, have been reported for foraminifera and also in other organismic groups such as macrofauna or bacteria (Pascal et al., 2008b, a; Quijón et al., 2008; Mayor et al., 2012). Fast incorporation of food pulses seems to be necessary to cover the high energy demand for reproduction and growth, since food concentrations (*Dunaliella sp.*) of less than 112 cells mm2 ( 0.40 $\mu$g C cm$^{-2}$ (Heinz et al., 2002)) do not permit growth or reproduction in *A. tepida*, while growth

rate and reproductive activity increase with additional food (Bradshaw, 1955, 1957). Lee et al. (1966) stated, that *Ammonia beccarii* specimen preferred to feed on 'new'(Fig 1. 10 days old) cultures of living *Chlorococcum* sp. over 'old' (up to 40 days old) cultures. Following these observations, a food limitation resulting from the aging of the phytodetrital food source could be present in this study. This proceeding degradation of phytodetritus appears to be particularly problematic to *A. tepida* specimen, which showed an increasing fraction of deceased individuals with progressing time (Fig 2). On the other hand, the



health condition of the specimen could be a response to the unnatural conditions of the laboratory cultivation. Compared to a previous feeding experiment by Linshy et al. (2014) with *A. tepida* fed with *D. tertiolecta* phytodetritus (614 mg C m$^{-2}$), uptake rates of carbon of 1899 pg ind$^{-1}$ h$^{-1}$ were much higher in this study after 48h than in the other feeding experiment (149 pg ind$^{-1}$ h$^{-1}$). This might be related to general differences in feeding of the tested strains.

The low affinity of *H. germanica* to *D. tertiolecta* phytodetritus could be explained by a generally lower carbon demand, or more likely a specialization on other food sources, for instance diatoms. The latter assumption corresponds to the preference of a diatom diet over sewage derived POM (Ward et al., 2003), the presence of a diatom cracking mechanism (Austin et al., 2005), the sequestration of chloroplasts derived exclusively from diatoms (Knight and Mantoura, 1985; Pillet et al., 2011; Cevasco et al., 2015), or the correlation of the distribution of *H. germanica* populations with high abundances of *Nitzschia*

sp. (Hohenegger et al., 1989). This coherence with food availability or organic matter accumulation respectively, was used to explain population distributions of a *H. germanica*/A. beccarii assemblages in muddy sediments of Spain (Papaspyrou et al., 2013; Diz et al., 2012). The different patterns of pC accumulation over time in the two species also reflect a different feeding behaviour. The onset of declining health conditions in *H. germanica* specimen at the end of the experiment (Fig 1.) was likely caused by starvation, due to the absence of an appropriate food source. These data report varying strategies in

phytodetritus feeding by means of source exploitation and time dependence. This can be useful for interpretation or explanation of successions in species dominances by means of food source availability and complements data from field surveys.

## 5   Conclusions

According to the results of this study, *H. germanica* exhibited a more vulnerable carbon retention behaviour to variations in temperature and a relatively low affinity to *D. tertiolecta* phytodetritus, suggesting other food preferences. *Ammonia tepida*

showed a broader tolerance range of temperature concerning carbon and nitrogen uptake and a highly effective food exploitation. The different response of the two species to the applied treatments implies different strategies and environmental adaptations. Consequently, a temperature related shift in abundances of the two species could alter carbon and nitrogen fluxes in intertidal sediments, with respect to massive chlorophyte detritus uptake of *A. tepida* and temperature sensitivity of *H. germanica*. In contrast to food derived carbon, the increasing accumulation of nitrogen was barely affected by temperature in

either species. This raises the hypothesis, that rapid temperature increases in *H. germanica* dominated sediments could cause a shift in organic carbon and nitrogen cycling towards enhanced nitrogen retention and carbon losses. A general enrichment in cytoplasmatic nitrogen in both species prove, that nitrogen has a higher potential to retain within the foraminiferal consumer. Further, increasing time had a prolonged negative influence on the food uptake rates from a single food pulse. A reduced availability of fresh phytodetritus sources is likely to cause such effects and appeared to be specifically limiting for *A. tepida*.

*Author contributions.* EAJ, PH and JW planned the study. EAJ and JW conducted the fieldwork and processed the samples in the lab. MW performed elemental and isotope analysis. JW performed data anlysis and wrote the manuscript. EAJ, PH and WW contributed by discussing results and critical revision of manuscript drafts.





*Competing interests.* The authors declare no competing interests.

*Acknowledgements.* Gerhard Schmiedl (Universität Hamburg) thankfully provided equipment for field sampling. Yvonne milker (Universität Leipzig) and Katharina Müller-Navarrra (Universität Hamburg) supported us at the sampling location. Patrick Bukenberger helped with lab work and microscoping at the University of Vienna.



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



**Table 2.** Effects of time and temperature. Results of a two-way ANOVA displaying the main effects time (sampling day) and temperature on the dependent variables (pC, pN, C, N) in *A. tepida* and *H. germanica* and their interactions (* p ≤ 0.050, ** p ≤ 0.010, p ≤ 0.001). Results for within group testing see text.

|  |  |  | Df | SM | F value | p value |  |
|---|---|---|---|---|---|---|---|
| *A. tepida* | pC | temperature | 2 | 17.341 | 7.898 | 0.002 | ** |
|  |  | day | 3 | 5.298 | 2.413 | 0.092 |  |
|  |  | temperature x day | 6 | 1.299 | 0.592 | 0.734 |  |
|  |  | Error | 24 | 2.196 |  |  |  |
|  | pN | temperature | 2 | 0.728 | 9.284 | 0.001 | ** |
|  |  | day | 3 | 1.939 | 24.716 | < 0.001 | *** |
|  |  | temperature x day | 6 | 0.084 | 1.073 | 0.406 |  |
|  |  | Error | 24 | 0.079 |  |  |  |
|  | C | temperature | 2 | 81.230 | 2.373 | 0.110 |  |
|  |  | day | 4 | 445.060 | 12.998 | < 0.001 | *** |
|  |  | temperature x day | 8 | 42.650 | 1.246 | 0.308 |  |
|  |  | Error | 30 | 34.240 |  |  |  |
|  | N | temperature | 2 | 2.112 | 2.848 | 0.074 |  |
|  |  | day | 4 | 4.221 | 5.690 | 0.001 | ** |
|  |  | temperature x day | 8 | 0.601 | 0.810 | 0.600 |  |
|  |  | Error | 30 | 0.742 |  |  |  |
| *H. germanica* | pC | temperature | 2 | 0.296 | 81.300 | < 0.001 | *** |
|  |  | day | 3 | 0.010 | 2.654 | 0.071 |  |
|  |  | temperature x day | 6 | 0.010 | 2.826 | 0.032 | * |
|  |  | Error | 24 | 0.004 |  |  |  |
|  | pN | temperature | 2 | 0.002 | 8.524 | 0.002 | ** |
|  |  | day | 3 | 0.027 | 133.994 | < 0.001 | *** |
|  |  | temperature x day | 6 | 0.001 | 5.260 | 0.001 | ** |
|  |  | Error | 24 | < 0.000 |  |  |  |
|  | C | temperature | 2 | 14.973 | 3.630 | 0.039 | * |
|  |  | day | 4 | 40.819 | 9.898 | < 0.001 | *** |
|  |  | temperature x day | 8 | 17.057 | 9.898 | 0.002 | ** |
|  |  | Error | 30 | 4.124 |  |  |  |
|  | N | temperature | 2 | 2.240 | 13.722 | < 0.001 | *** |
|  |  | day | 4 | 2.978 | 18.248 | < 0.001 | *** |
|  |  | temperature x day | 8 | 0.275 | 1.685 | 0.143 |  |
|  |  | Error | 30 | 0.163 |  |  |  |



**Table 3.** Carbon uptake rates. Regression variables, correlation coefficient $R^2$, standard error of estimate S and decay b the exponential decrease of carbon uptake rates of *H. germanica* and *A. tepida* versus time. Uptake in *H. germanica* decreases faster at higher temperatures, *A. tepida* shows a more steady decrease of the rates

|  |  | $R^2$ | $S$ | $b$ |
|---|---|---|---|---|
| *A. tepida* | 20°C | 0.873 | 11.382 | -0.262 ($\pm$ 0.048) |
|  | 25°C | 0.595 | 21.247 | -0.130 ($\pm$ 0.047) |
|  | 30°C | 0.918 | 4.797 | -0.130 ($\pm$ 0.017) |
|  | all T | 0.628 | 17.048 | - 0.162 ($\pm$ 0.030) |
| *H. germanica* | 20°C | 0.945 | 0.957 | -0.247 ($\pm$ 0.030) |
|  | 25°C | 0.904 | 0.797 | -0.343 ($\pm$ 0.058) |
|  | 30°C | 0.912 | 0.789 | -0.523 ($\pm$ 0.090) |
|  | all T | 0.732 | 1.680 | -0.316 ($\pm$ 0.053) |

**Table 4.** C:N ratios of algae and foraminiferal cytoplasm. Brackets contain standard deviation. Letters next to C:N values show Tukey Grouping of differences between sampling days within the respective temperature range of the species. Ratios in bold letters denote differences of C:N between *A. tepida* and *H. germanic* (Tukey HSD, p$\leq$ 0.005).

|  |  | 20°C | 25°C | 30°C |
|---|---|---|---|---|
| *D. tertiolecta* |  | 6.08 | - | - |
| *A. tepida* | 0 days | 4.20($\pm$0.42) | 4.20 ($\pm$0.42) **a** | 4.20 ($\pm$0.42) |
|  | 2 days | 5.32 ($\pm$0.59) | 5.32 ($\pm$0.20) **b** | 5.79 ($\pm$0.90) |
|  | 4 days | 5.02 ($\pm$0.21) | 4.97 ($\pm$0.07) **ba** | 5.70 ($\pm$0.36) |
|  | 7 days | 5.30 ($\pm$0.49) | 5.60 ($\pm$0.30) **b** | 4.99 ($\pm$0.05) |
|  | 14 days | 4.29 ($\pm$0.87) | 4.04 ($\pm$0.44) **a** | 4.41 ($\pm$0.35) |
| *H. germanica* | 0 days | 5.07 ($\pm$0.31) | 5.07 ($\pm$0.31) | 5.07 ($\pm$0.31)**b** |
|  | 2 days | 4.77 ($\pm$0.14) | 4.85 ($\pm$0.21) | 4.68 ($\pm$0.15) **b** |
|  | 4 days | 4.63 ($\pm$0.43) | 4.39 ($\pm$0.32) | 3.65 ($\pm$0.38) **a** |
|  | 7 days | 4.57 ($\pm$0.42) | 4.34 ($\pm$0.17) | 4.67 ($\pm$0.36) **b** |
|  | 14 days | 5.04 ($\pm$0.09) | 4.32 ($\pm$0.32) | 4.51 ($\pm$0.18) **ba** |