# Peer review of "Increased temperature causes different carbon and nitrogen processing patterns in two common intertidal foraminifera (*Ammonia tepida* and *Haynesina germanica*)"

_Biogeosciences, 2016_

## Referee Comment (RC1) · W. R. Hunter (Referee) · 5 Jan 2017

**Review of Wukovits et al. Effect of increased temperature on carbon and nitrogen uptake of two intertidal foraminifera (*Ammonia tepida* and *Haynesina germanica*).**

**General Comments**

The manuscript by Wukovits et al. describes a series of feeding experiments designed to test how warming affects the metabolism of two common coastal foraminifera. The experiments themselves are relatively well designed and provide useful information on how climate warming will affect the role of foraminifera in benthic carbon and nitrogen cycling. The paper represents a potentially important contribution to our understanding of foraminiferal physiology and their role in coastal biogeochemistry. I would, therefore, recommend its publication in Biogeosciences subject to minor revision. The paper is technically sound, although I think the authors need to clarify their statistical treatment of the data, which was rather confusing. A relatively large number of different tests where employed during data analysis and I believe this could be streamlined. There is quite a lot of discussion at present regarding the use of p-values, and I would urge the authors to reflect on this. They have an excellent dataset, and in many ways the results are evident even without recourse to inferential statistical tests. Thus simplifying the approach could be beneficial.

In places the paper is somewhat verbose, and also the authors are prone to using rather conversational language in their text. I would the authors to revise the manuscript to try and make the results in particular more concise and to adhere more strictly to the formal rules, which underlie academic writing. Particular attention needs to be paid to sentence structure to ensure the manuscript is logical and easily understood by the reader.

A number of methodological points also concern me. Firstly, I can find no justification for the range of temperatures at which these experiments where conducted. How do these compare with the temperatures typically recorded in temperature coastal sediments? This information is key to the study into a wider environmental context. Secondly, why did the authors use a 12 hour Light:dark cycle.

Given that these forams are heterotrophs living in the sediment is light likely to influence their activity, and if so how?

Aside from these points and the specific comments raised below, I would be happy to see this paper published in the near future.

**Specific Comments**

*Abstract*

Pg 1 Line 3-4: "Factors limiting the abundance of specific foraminiferal species can be temperature related stress tolerant or food source processing efficiency" – Firstly, this paper does not strictly discuss changes in the foraminiferal assemblage, but rather the physiological responses of two species to changes in temperature. Given that temperature is the key parameter that defines the physiology of all ectotherms I think this needs to be given considerably more weight.

Pg 1 Line 6: "phytodetritus ingestion" I would suggest that you are not specifically measuring ingestion, given the time-frame of the experiment. Instead "retention" given that you are measuring changes in 13C-enrichment over different time periods.

*Introduction*

Page 1 Line 20: replace "benthic species" with "benthic organisms".

Page 1 Line 21-23: "Along with future environmental changes…" This sentence is not particularly elegant, please revise and simplify the structure.

Page 2 Line 1: What do you mean by "competitive drawbacks?" Please Clarify.

Page 2 Line 4-2: "smaller benthic foraminifera can contribute up to 80 %..." revise this sentence along the following lines – smaller benthic foraminifera contribute up to 80 % of the protest biomass (refs!) and are an important component of the food web (refs!). In the current draft both "can" and "considered" are superfluous words. Also when writing a paper with a major biochemical / geochemical theme try not to use "element" in a non-chemical sense.

Page 2 Lines 23-24: "Research about coupling of food derived carbon…" Move both references to the end to read (Enge et al., 2014; 2016).

Page 3 Line 4: replaces "therefore aimed" with simply "aims to investigate…"

Page 3 Line 7-8: "This study also tests the question…" revise to "This study also tests which species shows…" Try not to overuse "therefore".

Page 3 Line 20-21: "This helps to interpret nutritional demands…" Please clarify this sentence.

Page 3 Line 22: "Studies aiming to develop new foraminiferal proxies…" would be a more elegant way to phrase this.

Page 3 Line 24: Replace "distinct observations" with "unique dataset".

*Methods*

Page 6 Line 20-23: Revise to "This test is recommended for sample sizes < 10 and is robust against heteroscedacity within the data (Moser and Stevens 1992; McDonald, 2014; Ruxon, 2006).

Page 6 Line 23-26: I am confused by the statistical methods described here. To compare within species differences you used a two-way ANOVA – why then did you then use either pairwise t-testing or a one-way ANOVA as post-hoc tests. Surely with the two-way ANOVA you can then use Tukey HSD or other post-hoc tests to test for the significant interactions, and the ANOVA reveals the significance of any independent effects? Please clarify this.

Page 7 Lines 1-4: I am not sure that you can design a biological experimental and assume any data distribution. You are correct that ANOVA is relatively robust against departures from normality. Heteroscedacity is, however, often an issue in biological experiments with small samples sizes and I would strongly advocate Zuur et al 's (2009 – Methods in Ecology and Evolution) approach to this, which calls for visual exploration of the data residuals. In any case this section of the text is rather poorly written, please revise and consider looing at how your data fits the assumptions of homoscedacity visually.

*Results*

Page 7 Line 13 (and elsewhere in the results): Delete "Remarkably" – restrict interpretive language to the discussion.

Page 7 Line 31: I believe you mean "Temperature effects cause…"

Page 9 Line 4: Delete "Same as for the $\delta^{13}C$ values" Poorly written sentence, please revise. Please concentrate on describing the results. It would be simpler to simply state the trend for the $\delta^{15}N$ values.

Page 10 Line 6: What do the authors mean by a diffuse fluctuation? This is not clear.

Page 10 Lines 7-8: Sentence is incomplete, how does *H. germanica* contrast with *A. tepida*.

Page 10 Line numbers do not appear to match up with the text.

Page 11 and 12. Line numbers do not match up with the text.

Page 11-12 Section 3.3. There is a lot of descriptive language here, which is quite confusing . The graphs probably provide a better summary of the data trends, please revise this section to make the trends clearer.

*Discussion*

Page 13. Line 3-5: What are the effects on *A. tepida*? The key findings of the paper are about the different responses to temperature between the two species. This needs to be highlighted early in the introduction.

Page 13. The line numbers do not match up again, there appear to be 2 line 5s?

Page 13 Line 5 (II). New paragraph "In general…"

Page 13. Why did you not control for microbial respiration within your experiment. You could have run a set of control incubations with the forams absent.

Page 14. Again the line numbering system is a mess.

Page 15 Line 24. "112 cells mm$^2$" Superscript missing.

Page 15 Line 27-28: The aging of the phytodetrital food source could have been controlled for within the experimental design. I think this may require further discussion. How does the phytodetritus quality change with aging?

Page 16. Line numbering is a again a problem here.

Page 16, 2$^{nd}$ paragraph: The author's discuss the low affinity of *H. germanica* to *D tertiolecta* as a food source. How representative is *D tertiolecta* as an algal food source. In intertidal sediments? diatoms represent the primary microalgal constituent of the microphytobenthos and coastal phytoplankton communities, wouldn't a diatom have provided a better POM source? Also could the production of MPB by the microphytobenthos not represent a major foram food source? I think these questions need to be addressed or alluded to.

Page 16, final paragraph: How do these results advance the potential use of forams as proxies for environmental monitoring. This is alluded to in the introduction and subsequently ignored.

---

## Referee Comment (RC2) · T. Toyofuku (Referee) · 14 Mar 2017

Review of "Effect of increased temperature on carbon and nitrogen uptake of two intertidal foraminifera (Ammonia tepida and Haynesina germanica)" by Julia Wukovits and others

Authors have tried to observe food uptake of two foraminiferal species from brackish water by laboratory feeding experiment with carbon and nitorogen isotopic laveling method. Though their experimental setup itself are not so novel, the method has been established to obtain the robust result. Even though the compound level isotope measurement was also possible to estimate the metabolic pathway, the current method is enough to observe uptake of nutrient into forainiferal cell. Some physical separation

among cell body, taken food material and its derivatives must be necessary to do such metabolism analyses. I think these will be future topic for authors.

This study succeed to show that the energy uptake and usage are variable between studied two species (Fig. 7). Double spike of carbon and nitrogen could efficiently clarify this difference. Authors' strategy is correctly functioning. I can identify this is the major finding of the study. The authors can emphasize this point with positive tone of writing.

All topics, the carbon and nitrogen circulation in the tidal flats, the energy dynamics by meiofauna and metabolism of the foraminifera, are included in the scope of BG and are also acceptable to the reader with great interesting. The study should be published in BG. I would like to recommend authors put some summarized numbers, e.g. carbon and nitrogen flux of both species, in abstract and conclusion for readers' convenient.

P1Title: The authors find the variable usage of nutrient with two species. I think authors can reflect this finding on the title to increase the impact.

P2L20 Such influence of bacteria can be estimated by a control condition without foraminifera. L25 "microalgae" Capitalize "m"

L35 "earth?s" Fix question mark.

P3L14 Could you see reproduction event during the course of experiment?

L31 Remove psu if you follow SI unit system. How about the deviations of Temperature and Salinity? Deviations are also necessary for other measured numbers.

P4L12 This parenthesis is not closed.

L20 Add "without sediment" if there are no sediment in culture system.

L25 Could you avoid hypoxia? Mention about the DO level even qualitatively.

L29 HgCl2, perhaps?

P5Fig1 You never measure with "unusual" individuals? Show the values if you have.

P5L6 Could you show the pictures of the individuals? The color of cytoplasm visually support to know foraminiferal uptake/digestion of algae.

P6L6 All C and N is directly transferred from algae to foraminifera? I expect some of them is transferred via other small organisms what also eat labeled algae. I would like to recommend authors describe such all possible path of uptake.

How do you think about the contribution of bacterial decomposition. Discussion can be done with isotopic composition of inorganic carbon in the cultural water.

P6L22 Close this parenthesis.

P7L9 Capitalize "s"

P7L21 Put "-" between 25°C and 30°C.

P10Fig 4 Why Ammonia's results are combined? Statistically identical?

P12Fig 7 A nice discovery. A. tepida just stored food in cytoplasm ? Degradation is rapid in H. germanica? This difference between species is not revealed without 15N labeling. I can identify this is one of the key result of this study. Could you support this difference by other observation (e.g. cytoplasmic streaming, pseudopodial activities)? Include the description of observation in Result and Discussion, if so.

P13L11 (actual 11) Foraminiferal flux can not explain this? I expect H. germania can quickly remineralization of carbon because the 15N:13C ratios show unproportional distribution. This may make 13C enrichment in DIC of water though the authors mentioned the influence of microbial activity. I also agree the bacterial influence, too. That would be proofed with control experiment without foraminifera in future study.

P13L27 I agree this opinion of authors. Could you observe the qualitative change of cellular volume under binocular? It is nicer if you show the pictures of individuals at 2nd and 4th day.

P15L14, 15 Italicize genus name.

P15L16 I agree the authors' consideration. I thought that the descriptions of ecological observation of foraminiferal individuals under binocular are valuable to support the consideration. The behaviors of pseudopodia, appearance of individuals, the color of soft tissue and others are really important to document with isotopic measurement.

P15L23 Be not italicized "sp".

P15L24 mm"2" "2" should be superscript.

P16L6 Italicize "A. beccarii"

P16L9 Remove "." after Fig 1.

---

## Author Comment (AC1) · 6 Apr 2017

Response to Reviewer #1:

JW: Reply, Julia Wukovits

The manuscript by Wukovits et al. describes a series of feeding experiments designed to test how warming affects the metabolism of two common coastal foraminifera. The experiments themselves are relatively well designed and provide useful information on how climate warming will affect the role of foraminifera in benthic carbon and nitrogen cycling. The paper represents a potentially important contribution to our understanding of foraminiferal physiology and their role in coastal biogeochemistry. I would, therefore, recommend its publication in Biogeosciences subject to minor revision. The paper is technically sound, although I think the authors need to clarify their statistical treatment of the data, which was rather confusing. A relatively large number of different tests where employed during data analysis and I believe this could be streamlined. There is quite a lot of discussion at present regarding the use of pvalues, and I would urge the authors to reflect on this. They have an excellent dataset, and in many ways the results are evident even without recourse to inferential statistical tests. Thus simplifying the approach could be beneficial. In places the paper is somewhat verbose, and also the authors are prone to using rather conversational language in their text. I would the authors to revise the manuscript to try and make the results in particular more concise and to adhere more strictly to the formal rules, which underlie academic writing. Particular attention needs to be paid to sentence structure to ensure the manuscript is logical and easily understood by the reader. A number of methodological points also concern me. Firstly, I can find no justification for the range of temperatures at which these experiments where conducted. How do these compare with the temperatures typically recorded in temperature coastal sediments? This information is key to the study into a wider environmental context. Secondly, why did the authors use a 12 hour Light:dark cycle.

> JW: Statistical analysis: The statistical data treatment was simplified by reducing inferential statistical tests. The two-way ANOVA was kept to depict an overview over the significance of time and temperature related effects on the measured variables. Post Hoc test were excluded, since the exploratory graphs clearly visualize the results.

> Temperature ranges: A justification for the chosen temperatures was given in the original manuscript, Page 3 Lines 13 – 16.

> 12 hour light:dark cycle:There are light triggerd processes in our sample species even though they are in general heterotrophic. Haynesina e.g. is known to harbour kleptoplasts derived from diatoms and to produce O2 (Jauffrais 2016). We wanted to keep the light conditions as close to natural condistions as possible maintain near natural metabolic activities of the specimens. Therefore we chose to conduct the feeding experiments with simulation of a diurnal day:night cycle.

> Manuscript language was improved.

Specific comments

Abstract

Page 1 Line 3 – 4: "Factors limiting the abundance of specific foraminiferal species can be temperature related stress tolerant or food source processing efficiency" – Firstly, this paper does not strictly discuss

changes in the foraminiferal assemblage, but rather the physiological responses of two species to changes in temperature. Given that temperature is the key parameter that defines the physiology of all ectotherms I think this needs to be given considerably more weight. Given that these forams are heterotrophs living in the sediment is light likely to influence their activity, and if so how? Aside from these points and the specific comments raised below, I would be happy to see this paper published in the near future.

> JW: This part of the abstract was adjusted according to Reviwer #1's recomendations: „Benthic foraminifera are highly abundant heterotrophic protists in marine sediments, but intertidal communities are expected to undergo future changes. Environmental changes can exceed the tolerance limits of intertidal species causing a shift in species composition which might result in altered nutrient fluxes. Factors limiting the abundance of specific foraminiferal species can be temperature related stress tolerance or food source processing efficiency." Was changed to: „Benthic foraminifera are highly abundant heterotrophic protists in marine sediments, but future environmental changes will challenge the tolerance limits of intertidal species. Metabolic rates and physiological processes in foraminifera are strongly depending on environmental temperatures. Temperature related stress could therefore impact foraminiferal food source processing efficiency and might result in altered nutrient fluxes through the intertidal food web."

Introduction

Page 1 Line 20: „benthic species" replaced with „benthic organisms"

Page 1 Line 21 – 23: „Along with future environmental changes…" This sentence is not particularly elegant, please revise and simplify the structure.

> JW: Revised: „Future environmental changes are expected to effect coastal foraminiferal communities and assemblage structures."

Page 2 Line 1: „Competitive drawbacks" replaced by „a lack of fitness"

Page 2 Line 4: "smaller benthic foraminifera can contribute up to 80 %..." revise this sentence along the following lines – smaller benthic foraminifera contribute up to 80 % of the protest biomass (refs!) and are an important component of the food web (refs!). In the current draft both "can" and considered" are superfluous words. Also when writing a paper with a major biochemical / geochemical theme try not to use "element" in a non-chemical sense.

> JW: Revised according to Reviewer.

Page 2 Line 33 – 34: "Research about coupling of food derived carbon…" Move both references to the end to read (Enge et al., 2014; 2016).

> JW: Revised according to Reviewer.

Page 3 Line 4: replaces "therefore aimed" with simply "aims to investigate…"

> JW: Revised according to Reviewer.

Page 3 Line 7 – 8: "This study also tests the question…" revise to "This study also tests which species shows…" Try not to overuse "therefore".

JW: Revised according to Reviewer.

Page 3 Line 20 – 21:"This helps to interpret nutritional demands…" Please clarify this sentence.

JW: Revised into: „This helps to interpret phytodetrital uptake in relation with foraminiferal carbon and nitrogen coupling…"

Page 3 Line 22: "Studies aiming to develop new foraminiferal proxies…" would be a more elegant way to phrase this.

JW: Revised according to Reviewer.

Page 3 Line 24: Replace "distinct observations" with "unique dataset"

JW: Revised according to Reviewer.

Material and Methods

Page 6 Line 20-23: Revise to "This test is recommended for sample sizes < 10 and is robust against heteroscedacity within the data (Moser and Stevens 1992; McDonald, 2014; Ruxon, 2006).

JW: Revised.

Page 6 Line 23 – 26: I am confused by the statistical methods described here. To compare within species differences you used a two-way ANOVA – why then did you then use either pairwise t-testing or a one-way ANOVA as post-hoc tests. Surely with the two-way ANOVA you can then use Tukey HSD or other post-hoc tests to test for the significant interactions, and the the ANOVA reveals the significance of anyindependent effects? Please clarify this.

JW: Statistical methods were reduced and unified. The two-way ANOVA shows significant effects within temperature and time treatments and interactive effects of time and temperature. Further post-hoc testing was excluded in the RM, since the exploratory graphs clearly visualize the impact of temperature on A. tepida and H. germanica.

Page 7 Lines 1 – 4: I am not sure that you can design a biological experimental and assume any data distribution. You are correct that ANOVA is relatively robust against departures from normality. Heteroscedacity is, however, often an issue in biological experiments with small samples sizes and I would strongly advocate Zuur et al 's (2009 – Methods in Ecology and Evolution) approach to this, which calls for visual exploration of the data residuals. In any case this section of the text is rather poorly written, please revise and consider looing at how your data fits the assumptions of homoscedacity visually.

JW: As recommended by the Reviewer, heteroscedascity of the variables was tested by exploration of the residuals after Zuur et al. 2009. The results are attached as figures and will be available as supplementary material for the publication.

The Methods section Page 6 Line 23 – Page 7 Line 7 were condensed and rewritten according to method adaptations and recomendations of Reviewer #1.

Results

Page 7 Line 13 (and elsewhere in the results): Delete "Remarkably" – restrict interpretive language to the discussion.

JW: Revised according to Reviewer.

Page 7 Line 31: I believe you mean "Temperature effects cause…"

JW: Revised.

Page 9 Line 4: Delete "Same as for the δ13C values" Poorly written sentence, please revise. Please concentrate on describing the results. It would be simpler to simply state the trend for the δ15N values.

JW: Revised according to Reviewer.

Page 10 Line 6 – 8: What do the authors mean by a diffuse fluctuation? This is not clear

JW: This section was rewritten: „Like in pC, time and temperature showed interactive effects on the pN content in H. germanica  (Table 2)

Page 10 Lines 7-8: Sentence is incomplete, how does H. germanica contrast with A. tepida.

JW: Revised: Cytoplasmatic N showed less variation with temperature in A. tepida than in H. germanica.

Page 11 Line 3.3: There is a lot of descriptive language here, which is quite confusing . The graphs probably provide a better summary of the data trends, please revise this section to make the trends clearer.

JW: Section was revised.

Discussion

Page 13. Line 3-5: What are the effects on A. tepida? The key findings of the paper are about the different responses to temperature between the two species. This needs to be highlighted early in the introduction.

JW: The initial part of the discussion was revised: „The metabolism of H. germanica was significantly affected by elevated temperatures. Higher temperatures reduced the amount of pC in the foraminiferal cytoplasm, while otherwise a slighter effect on pN indicates a lower impact of temperature on the general uptake of algal phytodetritus. In contrast, pC processing in A. tepida was favoured at the intermediate experimental temperature level of 25°C, demonstrating different optimum phytodetritus processing temperatures in the two species."

Page 13 Line 3 – 5: section was completed with: „In contrast, pC processing in \textit{A. tepida} was favoured at the intermediate experimental temperature level of 25°C, demonstrating different optimum phytodetritus processing temperatures in the two species."

Page 13 Line 5/II (OM): New paragraph after „In general…" . Revised.

Page 13: Why did you not control for microbial respiration within your experiment. You could have run a set of control incubations with the forams absent.

> JW: An aliquot of microbial respiration, responsible for the relatively high delta 13C values was most likely caused by bacteria introduced with the foraminiferal specimen. Foraminifera were cleaned with brushes prior to transfer to the experimental dishes, but some bacteria were most likely still attached to the foraminiferal test. The synthetic seawater was filtered and the food source was supposed to be sterile. So calculation of foraminiferal caused increase in delta 13C values would have been not very accurate, even when subtracting from a foraminifera free control group.

Page 15 Line 24: „112 cells mm2" superscript „2", revised.

Page 15 Line 27 – 28: The aging of the phytodetrital food source could have been controlled for within the experimental design. I think this may require further discussion. How does the phytodetritus quality change with aging?

> JW: Methods for counting of bacterial colonisation of the detritus (e.g. Dapi-Staining/fluorescence microscopy, molecular biological methods...) were not available at the time the experiment was carried out. The authors will consider filtration of algal residue after the collection of specimen at the end of the experiment, to carry out additional EA and IRMS analysis. Comments on aging algae material were added to the discussion:

> „Aging or phytodetrital degeneration and its change in quality in marine benthic environments is a result of bacterial colonisation and subsequent transformation and mineralization of the algal material (Bühring et al. 2006, Gihring et al. 2009, Middleburg et al. 2000, Moodley et al. 2002). Through bacterial mineralization, the degrading detritus decreases in quality with an increase in C:nutrient ratios. However, microorganisms colonizing patches of degrading detritus are likely to be incorporated simultanously with foraminiferal phytodetritus grazing. Therefore, a fraction of the isotope label within the foraminiferal cytoplasm could be a result of indirect label intake with phytodetritus associated microorganisms especially towards the end of the experiment."

Page 16 2nd Paragraph: The author's discuss the low affinity of H. germanica to D. tertiolecta as a food source. How representative is D. tertiolecta as an algal food source. In intertidal sediments? diatoms represent the primary microalgal constituent of the microphytobenthos and coastal phytoplankton communities, wouldn't a diatom have provided a better POM source? Also could the production of MPB by the microphytobenthos not represent a major foram food source? I think these questions need to be addressed or alluded to.
Page 16, final paragraph: How do these results advance the potential use of forams as proxies for environmental monitoring. This is alluded to in the introduction and subsequently ignored.

> JW: Final paragraph revised: „In general, quality of organic carbon or foraminiferal food sources oscillates throughout the year and includes allochtonous detritus (Heip 1995), while the main source of primary production in intertidal environments are microphytobenthic diatoms, accompanied by chlorophytes or other autotrophic microorganism, seasonally suppressing diatom dominance (Scholz & Liebezeit 2012). In this experiment, D. tertioloecta was used as food source, because this species is easy to maintain in culture and was previously used in several

feeding experiments with benthic foraminifera and serves as a representative for allochtonous detrital carbon sources. Ratios of A. tepida and H. germanica abundances are considered to be related to organic matter quality or environmental variability, due to their different feeding specialisations or environmental adaptations (DeNooijer 2007). High amounts of pC (max. 30 % pC:C) in A. tepida compared to H. germanica (max. 2.6 % pC:C), together with the low influence on temperature on the feeding behavior of A. tepida, prove an opportunistic feeding behavior and generalist temperature adaptations in A. tepida. These findings will help to interpreted oscillations in abundances of these two common intertidal foraminiferal species in relation with organic carbon quality and environmental temperatures."

---

## Author Comment (AC2) · 6 Apr 2017

Response to Reviewer #2:

Authors have tried to observe food uptake of two foraminiferal species from brackish water by laboratory feeding experiment with carbon and nitorogen isotopic laveling method. Though their experimental setup itself are not so novel, the method has been established to obtain the robust result. Even though the compound level isotope measurement was also possible to estimate the metabolic pathway, the current method is enough to observe uptake of nutrient into forainiferal cell. Some physical separation among cell body, taken food material and its derivatives must be necessary to do such metabolism analyses. I think these will be future topic for authors. This study succeed to show that the energy uptake and usage are variable between studied two species (Fig. 7). Double spike of carbon and nitrogen could efficiently clarify this difference. Authors' strategy is correctly functioning. I can identify this is the major finding of the study. The authors can emphasize this point with positive tone of writing. All topics, the carbon and nitrogen circulation in the tidal flats, the energy dynamics by meiofauna and metabolism of the foraminifera, are included in the scope of BG and are also acceptable to the reader with great interesting. The study should be published in BG. I would like to recommend authors put some summarized numbers, e.g. carbon and nitrogen flux of both species, in abstract and conclusion for readers' convenient.

> JW: Maximum individual values of individual carbon and nitrogen uptake per species were summarized in the conclusion of the revised manuscript.

P1Title: The authors find the variable usage of nutrient with two species. I think authors can reflect this finding on the title to increase the impact.

> JW: Title was changed: „Increased temperature causes different carbon and nitrogen processing patterns in two common intertidal foraminifera (*Ammonia tepida* and *Haynesina germanica*)."

Page 2 Line 20: Such influence of bacteria can be estimated by a control condition without foraminifera.

> JW: see response to Reviewer #1.

Page 2 Line 25: L25 "microalgae" Capitalize "m"

> JW: Revised.

Page 2 Line 35: "earth?s" Fix question mark.

> JW: Revised.

Page 3 Line 1: Could you see reproduction event during the course of experiment?

> JW: No, there was no reproduction within the course of the experiment. „The temperatures chosen for this experiment correspond to experimentally determined values that cover optimum or tolerance ranges for growth and reproduction in laboratory cultures of intertidal foraminifera (refs)" to „The temperatures chosen for this experiment correspond to experimentally determined values that cover optimum or tolerance ranges of physiological processes in intertidal foraminifera (refs)." To avoid confusion about observed factors in this study.

Page 4 Line 12: This parenthesis is not closed.

> JW: Revised.

Page 4 Line 25: Could you avoid hypoxia? Mention about the DO level even qualitatively.

> JW: Information about O2 levels was added: „...measured O2 at sampling days was 5 - 8 mg L -1"

Page 4 Line 29: HgCl2, perhaps? HgCl3 replaced by HgCl2

> JW: Revised. HgCl3 replaced by HgCl2.

Page 5 Fig. 1: You never measure with "unusual" individuals? Show the values if you have.

> JW: No, „unusual" individuals were never measured. They are still stored within our freezer. It would be indeed interesting, to measure those individuals for comparison. But this was not subject of this study.

Page 5 Line 6: Could you show the pictures of the individuals? The color of cytoplasm visually support to know foraminiferal uptake/digestion of algae.

> JW: Unfortunately, we do not have pictures of the actual measured individuals. My microscope at that time was not equipped with a camera and I focused on swift processing of picked or frozen individuals for analysis.

Page 6 Line 6: All C and N is directly transferred from algae to foraminifera? I expect some of them is transferred via other small organisms what also eat labeled algae. I would like to recommend authors describe such all possible path of uptake.

> JW: This consideration was included in the discussion.

Page 6 Line 22: Close this parenthesis.

> JW: Sentence changed according to suggestion of Reviewer #1.

Page 7 Line 9: Capitalize "s"

> JW: Revised.

Page 7 Line 21: Put "-" between 25∘C and 30∘C.

> JW: Revised.

Page 10 Fig 4: Why Ammonia's results are combined? Statistically identical?

> JW: Yes, A. tepida results are combined, because they do not differ significantly. A clarifying comment was added to the Figure description.

Page 12 Fig 7 & Page 13 Line 27: A nice discovery. A. tepida just stored food in cytoplasm? Degradation is rapid in H. germanica? This difference between species is not revealed without 15N labeling. I can identify this is one of the key result of this study. Could you support this difference by other observation (e.g. cytoplasmic streaming, pseudopodial activities)? Include the description of observation in Result and Discussion, if so.

> JW: No, we did not carry out observations on cytoplasmatic streaming or pseudopodial activity or took pictures of the specimens after the end of the incubation period. All foraminifera in the experimental dishes were carefully collected, tranferred to Eppendorf(C) tubes and frozen for further cleaning and processing. The total number of individuals needed for EA and IRMS was

not clarified at the course of the experiment, so all 150/170 individuals per replicate were stored for EA and IRMS analysis. Observations on 2700 individuals per sampling day (to observe pseudopodial movement) would have exceeded personal and time capacity. We were taking into account to transfer all individuals with the same routine, to avoid deviations of the results e.g. due to prolongued processing (further observations under the microscope for pseudopodial activity etc.). But such observations are valuable for additional prove of the data. An optional observation of random samples / preservation of individuals for additional analysis within sample series will be taken into account for further studies.

Page 15 Line 14, 15: Italicize genus name.

JW: Revised.

Page 15 Line 16: Foraminiferal flux can not explain this? I expect H. germania can quickly remineralization of carbon because the 15N:13C ratios show unproportional distribution. This may make 13C enrichment in DIC of water though the authors mentioned the influence of microbial activity. I also agree the bacterial influence, too. That would be proofed with control experiment without foraminifera in future study.

JW: Reviewers considerations were included into the discussion: „Interestingly, there is no significant temperature effect on pC between 25°C and 30°C. This implicates a critical threshold for this species between 20°C and 25°C. Above this level, the mineralization of carbon increases, as the DIC-C rises at 25°C and 30°C as a result of elevated respiratory activity. The decoupling of pC and pN in H. germanica over time supports this observation."

About the control experiments and bacterial contamintation see answer to Reviewer #1's comment on Page 13.

Page 15 Line 23: Be not italicized "sp".

JW: Revised.

Page 15 Line 24: mm"2" "2" should be superscript.

JW: Revised.

Page 16 Line 6: Italicize "A. beccarii".

JW: Revised.

Page 16 Line 9: Remove "." after Fig 1.

JW: Revised.